# HM-ANN: Efficient Billion-Point Nearest Neighbor Search on Heterogeneous Memory

**Jie Ren**
University of California, Merced
jren6@ucmerced.edu

**Minjia Zhang**
Microsoft Research
minjiaz@microsoft.com

**Dong Li**
University of California, Merced
dli35@ucmerced.edu

## Abstract

The state-of-the-art approximate nearest neighbor search (ANNS) algorithms face a fundamental tradeoff between query latency and accuracy, because of small main memory capacity: To store indices in main memory for fast query response, They have to limit the number of data points or store compressed vectors, which hurts search accuracy. The emergence of heterogeneous memory (HM) brings opportunities to largely increase memory capacity and break the above tradeoff: Using HM, billions of data points can be placed in main memory on a single machine without using any data compression. However, HM consists of both fast (but small) memory and slow (but large) memory, and using HM inappropriately slows down query time significantly. In this work, we present a novel graph-based similarity search algorithm called HM-ANN, which takes both memory and data heterogeneity into consideration and enables billion-scale similarity search on a single node without using compression. On two billion-sized datasets BIGANN and DEEP1B, HM-ANN outperforms state-of-the-art compression-based solutions such as L&C [13] and IMI+OPQ [12] in recall-vs-latency by a large margin, obtaining 46% higher recall under the same search latency. We also extend existing graph-based methods such as HNSW and NSG with two strong baseline implementations on HM. At billion-point scale, HM-ANN is 2X and 5.8X faster than our HNSW and NSG baselines respectively to reach the same accuracy.

## 1 Introduction

Efficient billion-scale nearest neighbor search has become a significant research problem [6, 7, 22, 23], inspired by the needs of machine learning based applications. Since the number of entities (images, documents, etc) grows enormously fast, it becomes challenging to find correspondences in large datasets when there is a requirement for real-time responses (e.g., in several milliseconds). Exhaustive search is infeasible at billion-point scales, because it is extremely computational demanding. Hence, practitioners resort to indexing structures that perform the approximate nearest neighbor search (ANNS) by restricting a query to search only a subset of the dataset that includes the desired neighbors [10, 19, 25]. Among those ANNS, it has been demonstrated that similarity graphs, such as Hierarchical Navigable Small World (HNSW) [29] and Navigating Spread-out Graph (NSG) [16], obtain superior performance relative to tree structure based [9, 10, 31, 46], locality sensitive hashing (LSH) based [18], and inverted multi-index (IMI) based [25] approaches, and they overall provide the best-in-class latency-vs-accuracy trade-off on most public benchmark datasets.

While obtaining good search speed and accuracy, one major limitation of existing similarity graphs is that they are very memory consuming and easily run out of memory with a few hundred millions of vectors. When the dataset becomes too large to fit on a single machine, the compressed representations of the database points are used, such as Hamming codes [33] and product quantization [13, 17, 21, 24, 32]. However, the performance of these methods deteriorates rapidly at higher recall targets,

because they calculate approximate distance based on compressed vectors instead of on the original data vectors. Douze et. al. [13] propose Link-and-Code (*L&C*), which combines a similarity graph with quantized nodes and exploits neighbor nodes to refine the estimation of distance. However, this approach still works poorly at high recall targets. In [40], the authors explore slow storage to achieve billion-scale ANNS in a single machine. However, this approach is based on a fundamental assumption that the persistent media such as SSD is several orders of magnitude slower than DRAM. Based on this assumption, data accesses to the persistent media during search should be zero. As a result, it maintains a copy of compressed data in memory with product quantization [40], which results in loss of in-memory search quality. It then preforms a re-ranking using full-precision coordinates stored on SSD, using block-level data accesses but with expensive SSD accessing time.

In this work, we present a fast and accurate approximate nearest neighbor search algorithm for extremely large scale ANN search, called HM-ANN, which is built on top of Heterogeneous Memory. Heterogeneous Memory (HM) combines cheap, slow but extremely large memory with expensive, fast but small memory (e.g., traditional DRAM) to achieve a good balance between production cost, memory performance and capacity. The emergence of HM brings opportunities to significantly improve ANNS. Because of the large memory capacity, HM can use full-precision vectors with accurate distance computation. Since memory access latency/bandwidth of slow memory component in HM is much faster than slow storage such as SSD, it is possible to occasionally access data in slow memory during search without paying expensive cost of data accesses. That being said, releasing full performance potential of HM for ANNS is challenging. Although the slow memory such as PMM performs ∼80X times faster than SSD, it is still ∼3X slower than DRAM in terms of random access latency. Therefore, a naive data placement strategy can hurt the search efficiency badly. It then raises the following research question: can we leverage HM for ANNS to achieve both high search accuracy and low search latency, especially when the dataset cannot in DRAM (fast memory)? Specifically, the algorithm should have a clear advantage over the state-of-the-art ANNS solutions.

HM-ANN enables fast and highly accurate billion-scale ANNS on HM. In particular, we make the following contributions. (1) We present a fast and accurate billion-scale nearest neighbor search solution on a single node without compression. Specially, we generalize the HNSW construction algorithm to have a top-down insertion phase and a bottom-up promotion phase. The top-down phase creates navigable small world graph as the bottom-most layer, which is also the largest, placed to the slow memory; The bottom-up promotion phase promotes pivot points from the bottom layer graph to form upper layers that are placed in the fast memory, which allows most search accesses to happen in fast memory without losing much accuracy. (2) We explore memory management techniques such as dynamic migration to prefetch to-be-accessed data from slow memory to fast memory and parallel search to reduce search time in slow memory. (3) We introduce a performance model to select search-related hyperparameters that satisfy search time and recall constraints. (4) We conduct extensive evaluation and show that on two billion-scale datasets, HM-ANN provides 95% top-1 recall in less than one millisecond; HM-ANN outperforms state-of-the-art compression-based solutions such as L&C [13] and IMI+OPQ [12] in terms of recall-vs-latency by a large margin, getting 46% higher recall under the same search latency budget; Since NSG and HNSW have never been scaled up to a billion vector on a single machine, we create two strong baselines for them: using first-touch NUMA and hardware-managed caching, respectively. Our results show that for 95% top-1 recall, HM-ANN outperforms the baselines by 2X-5.8X in terms of search latency.

## 2 Preliminary and Related Works

### 2.1 ANNS and Similarity Graphs

Similarity graphs like HNSW [29] and NSG [16] have demonstrated superior performance with polylogarithmic search and graph construction complexity for ANNS [15, 26, 40]. Take HNSW as an example, which consists of multiple layers. The bottom-layer (L0) contains all database elements, and the above layers are randomly selected, nested subsets of database elements. The sizes of the layers follow a geometric progression. During the graph construction phase, HNSW connects elements in each layer based on the closeness relationship. The connections of an element consist both long-range links and short-range links to establish the small world properties. HNSW constrains the length of the neighbors list of each element by a parameter $M$. HNSW starts the search at the top layer, and performs a *1-greedy search* until it reaches the nearest neighbor of the query in that layer. That node is then used as an entry point in the next layer to start search again. At the bottom layer L0, which

contains all elements, HNSW performs a *best-first beam search* to get the final candidates. HNSW uses a parameter $efSearch$, which decides the candidate queue length, to control search time vs. accuracy trade-off. Despite their outstanding performance, similarity graphs are memory-consuming. For example, for the Deep1B [6] dataset, they require 384 bytes per vector, which translates to >350 GB DRAM when including all overheads of data structures, causing out-of-memory failure. Therefore, existing work mostly evaluate their solutions with a few millions vectors [16, 29].

## 2.2 Heterogeneous Memory

Heterogeneous memory (HM) is emerging. It combines multiple memory components to construct main memory. HM is typically composed of a high-capacity memory technology such as non-volatile memory (but high memory access latency) and a high-performance memory technology (with limited memory capacity) such as DRAM. To make HM performance close to that of DRAM-only, previous work focuses on hardware- [3, 11, 35, 36, 41] and software-based [14, 27, 43, 45, 44, 28, 37] solutions to manage data placement on HM. Optane PMM and DRAM are commonly used to build HM. With PMM, the memory capacity on a single machine can achieve 6TB [20]. However, the latency and bandwidth of PMM is only 1/3 and 1/6 of DRAM. There are two operating modes for PMM, *Memory Mode* and *App-direct Mode*. In Memory Mode, DRAM works as a hardware-managed cache to PMM. Running the application in this mode does not require application modifications. App-direct Mode allows the programmer to explicitly control memory accesses to PMM and DRAM. HM-ANN works in App-direct Mode and outperforms Memory Mode in billion-scale dataset search (Section 4).

## 3  HM-ANN

The design of HM-ANN generalizes HNSW, whose hierarchical structure naturally fits into HM. Elements in upper layers consume a small portion of the memory, making them good candidates to be placed in fast memory (small capacity); The bottom-most layer has all the elements and has the largest memory consumption, which makes it suitable to be placed in slow memory. Unlike HNSW, where the majority of search happens in the bottom-most layer, elements in upper layers now have faster access speed, so it is a reasonable strategy to increase the access frequency of upper layers. On the other hand, since accessing L0 is slower, it is preferable to have only a small portion of it to be accessed by each query. The key idea of HM-ANN is therefore to build high-quality upper layers and make most memory accesses happen in fast memory, in order to provide better navigation for search at L0 and reduce memory accesses in slow memory.

**Notations.** In the rest of the paper, we let $V$ denote the dataset with $N = |V|$ to build the graph; we refer the graph in the layer $i \in \{0, 1, ..., l\}$ of HM-ANN as $G_i = (V_i, E_i)$ where $V_i$ is the vertex set and $E_i$ is the edge set. We refer $N_i$ as the number of elements in the layer $i$, and we have $N_i = |V_i|$. Because L0 contains all the elements in database, we have $V_0 = V$ and $N_0 = N$. Based on the hierarchical structure of HM-ANN, we have $V_i \subsetneq V_{i-1}$. Similar to the existing effort [29], we introduce $M_i$ as the maximum number of established connection for each point $v$ in the layer $i$. For $v \in V$, we let $D(v)$ denote the degree of node $v$, and $D(v) = \sum_{u \in V} m(v, u)$ where $m(v, u) = 1$ if there exits a link between node $v$ and node $u$.

### 3.1  Graph Construction via Top-Down Insertions and Bottom-up Promotions

We generalize the HNSW construction algorithm to include two phases: a top-down insertion phase and a bottom-up promotion phase (Alg. 1).

**Top-down insertions.** The top-down insertion phase is the same as HNSW (Line 1 in Algorithm 1), where we incrementally build a hierarchical graph by iteratively inserting each vector $v$ in $V$ as a node in $G$. Each node will generate up to $M$ (i.e., the neighbor degree) out-going edges. Among those, $M - 1$ are short-range edges, which connect $v$ to its $M - 1$ nearest neighbors according to their pair-wise Euclidean distance to $v$. The rest is a long-range edge that connects $v$ to a randomly picked node, which may connect other isolated clusters. It is theoretically justified that graphs (e.g., L0) constructed by inserting these two types of edges guarantees to have the small world properties [16, 42, 29].

**Bottom-up promotions.** The goal of the second phase is to build a high-quality projection of L0 elements into the layer 1 (L1), such that search in L0 can find true nearest neighbours of the query with only a few number of hops. Ideally, HM-ANN wants to achieve the goal that performing

1-greedy search in L0 is sufficient to achieve high recall, so that the slowdown caused by accessing the slow memory is minimal. A straightforward way to project the L0 elements into L1 is to randomly select a subset of elements in L0 to be L1, similar to what HNSW already does to build upper layers. However, we observe that such an approach leads to poor index quality. As a result, many searches end up happening in L0 (slow memory), causing long search latency.

---

**Algorithm 1: HM-ANN Graph Construction Algorithm.**

---

**Input:** vector set $V$, vector dimension $d$, number of established connection $M$, size of dynamic candidate list $efConstruction$
**Output:** Multi-layer graph HM-ANN
**Parameters:** # of nodes in layer $i$ $N_i$, HM-ANN layer depth $l$

1    build graph $hnsw \leftarrow HNSW(V, d, M, efConstruction)$ ;
2    **for** $v$ *in* $V$ **do**
3        $D[v] \leftarrow$ the degree of $v$ as in zero-layer L0;
4    sort $D$ for descending order ;
5    remove nodes in layer 1 to $l$ ;
6    $ep \leftarrow$ get the highest degree node $v$ in $D(v)$ ;
7    **for** $v$ *in* $V$ *in* $D(v)$ *descending order* **do**
8        **for** $i \leftarrow l...1$ **do**
9            **if** $N_i == 0$ **then**
                `// layer i is full`
10              $W \leftarrow$ search_layer$(v, \{ep\}, ef = 1, i)$;
11              $ep \leftarrow$ get nearest vector from $W$ to $v$;
12            **else**
                `// add v in layer i to 1`
13              **for** $j \leftarrow i...1$ **do**
14                  $W \leftarrow$ search_layer$(v, \{ep\}, efConstraction, j)$ ;
15                  $neighbors \leftarrow$ heuristic select $M_i$ nodes from $W$ in layer $j$ ;
16                  add bidirectional connections from $neighbors$ to $v$ at layer $j$;
17                  shrink connections if $\exists q \in neighbor$ and $D_{out}(q) > M_i$;
18                  $N_j = N_j - 1$;
19            beark;

---

HM-ANN uses a *high-degree promotion strategy* (Lines 7-19 in Algorithm 1). This strategy promotes elements with the highest degree in L0 into L1. From the layer $i$ ($i \geq 2$) to $i + 1$, HM-ANN promotes high-degree nodes to upper layer with a promotion rate of $1/M$, where $M$ is the maximum number of neighbors for each element (i.e., $M_i = M$, where $i = 2...l$). The similar promotion rate setting is used in HNSW [29] and typical skip list [34].

HM-ANN increases search quality in $L1$ by promoting more nodes from $L0$ to $L1$ and setting the maximum number of neighbors for each element in L1 to $2 \times M$ (i.e., $M_1 = 2 \times M$). The number of nodes in upper layers ($N_i$, where $i = 1..l$) is decided by available fast memory space. Excluding the fast memory space for dynamic migration (discussed in Section 3.2) and data structure used for search (e.g, the visited elements set $VE$ in Algorithm 2), the remaining fast memory space is used for storing data and links for each node. Section 3.4 quantifies memory usage in each layer, from which we can calculate $N_i$ for each layer.

The high-degree promotion strategy is based on the following observation. The hub nodes of the graph at L0 are those nodes with a large number of connections (i.e., high degree). In the small world navigation algorithm, a higher degree node provides better navigability [8]. Most of the shortest paths between nodes flow through hubs. In other words, the average length of the navigation path (i.e., number of hops) is the smallest, when the adjacent node with the highest degree is selected as the next hop. By promoting the high-degree nodes, the resulting L1 layer allows HM-ANN to effectively reduce the number of search in L0, compared with the random promotion strategy.

## 3.2 HM-ANN Graph Search Algorithm

**Fast memory search.** The search in fast memory begins at the entry point in the top layer and then performs 1-greedy search from the top layer to the layer 2, which is the same as in HNSW. To narrow down the search space in L0, HM-ANN performs the search in L1 with a search budget controlled by $efSearch_{L1}$ by using Algorithm 2. $efSearch_{L1}$ defines the size of dynamic candidate list in

L1. Those candidates in the list are used as entry points for search in L0 (HNSW uses just one entry point), in order to improve search quality in L0. We provides algorithm details in the appendix.

**Parallel L0 search.** In L0, HM-ANN evenly partitions the candidates from searching L1 and uses them as entry points to perform *parallel multi-start 1-greedy search* with $Thr$ threads in parallel as shown in Algorithm 2. The top candidates from each search are collected to find the best candidates. Parallel search makes best use of memory bandwidth and improves search quality without increasing search time. $Thr$ is determined by peak memory bandwidth constrained by hardware divided by memory bandwidth consumption by one thread, which is easy to calculate.

---

**Algorithm 2: HM-ANN Search Layer**

---

**Input:** query vector $q$, enter points set $EP$, number of nearest neighbors to query $q$ to return $ef$, layer number $l$
**Output:** $ef$ nearest vectors to $q$
**Parameters:** # of threads $Thr$, set of visited elements VE, set of candidates $C$, dynamic list of found nearest neighbors $W$

1  $Thr = min(Thr, |EP|)$
2  partition $EP$ into $EP_i, i \leftarrow Thr - 1...0$
3  **do in parallel**
4  $\quad VE_t \leftarrow EP$ ; $C_t \leftarrow EP_t$; $W_t \leftarrow EP$
5  $\quad$ **while** $|C_t| > 0$ **do**
6  $\quad\quad$ **if** *min_dist(q,$C_t$)>max_dist(q,$W_t$)* **then**
7  $\quad\quad\quad$ break;
8  $\quad\quad$ evaluate neighbors of $c \in C_t$
9  $\quad\quad$ update $VE_t$ and $W_t$
10  merge $W_i$ into $W$, $i \leftarrow Thr - 1...0$
11  **return** $ef$ nearest vectors from $W$ to $q$

---

Different from the SSD-based ANNS [40, 47], the data in slow memory in HM-ANN can be directly accessed by processors, and there is no duplication between fast and slow memories. However, due to high latency and low bandwidth of slow memory, HM-ANN should still make memory accesses in fast memory as many as possible. HM-ANN implements a software-managed cache in fast memory to prefetch data from slow memory to fast memory before the memory access happens. In particular, HM-ANN reserves a space in fast memory ($\sim$2 GB) called *migration space*. When searching L1, HM-ANN asynchronously copys neighbor elements of those candidates in $efSearch_{L1}$ and the neighbor elements' connections in L1 from slow memory to the migration space in fast memory. When the search in L0 happens, there is already a portion of to-be accessed data placed in fast memory, which leads to shorter query time.

## 3.3 Performance Model-Guided Parameter Selection

The overall search quality of HM-ANN is related to the choice of $efSearch$ at L1 (i.e., $efSearch_{L1}$) and $efSearch$ at L0 (i.e., $efSearch_{L0}$), which controls the number of distance computation happens in fast memory and slow memory, respectively. To achieve a low query latency, ideally we would like $efSearch_{L0}$ to be as small as possible, such as 1-greedy search ($efSearch_{L0} = 1$). However, although searching L1 narrows down the L0 search into a small local region, to have a high search quality requires that $efSearch_{L0}$ can not be too small, because the nearest neighbors not included L1 and are not visited in L0 are definitely lost. Given the large search space of $efSearch_{L1}$ and $efSearch_{L0}$, it is preferable to have a systematic way to do parameter selection. This section provides a performance model for HM-ANN, with an eye towards being able to set $efSearch_{L1}$ and $efSearch_{L0}$ properly to meet the goal of having low response time and high accuracy.

**Response time constraint.** To provide interactive service, the search latency must be lower than a response time limit. In HM-ANN, we model the search latency as $T = T_{L1*} + T_{L0}$, where $T_{L1*}$ models search time in L1 and above, which is primarily dominated by search in $L1$, and $T_{L0}$ models search time in L0. The average query time at a layer is bounded by $efSearch \times C \times T_{DC}$, where $efSearch$ is the size of dynamic candidate list in the layer and can be viewed as the beam length in the best-first beam search; $C$ is the average number of distance computations per beam before finding the nearest neighbor at a layer; $T_{DC}$ is the execution time to calculate a pair-wise distance.

$T_{DC}$ is a constant and can be measured offline on both fast memory ($T_{DC_{fast\_mem}}$) and slow memory ($T_{DC_{slow\_mem}}$). $C$ is calculated by $C = \#steps \times DC\_per\_step$, which is a multiplication of the average number of steps before we reach the nearest neighbor ($\#steps$) and maximum number of distance computation per step ($DC\_per\_step$). $\#steps$ in a layer is bounded by a constant [29] based on the theory of Delaunay graph and is independent of the dataset size; $DC\_per\_step$ is bounded by the maximal out-degree $M$. When modeling search time in L0, we consider the effect of parallel search with a parallel degree $Thr$. For the execution time, we therefore have:

$$T = T_{L1*} + T_{L0}$$

$$= efSearch_{L1} \times C \times T_{DC_{fast\_mem}} + \left\lceil \frac{efSearch_{L1}}{Thr} \right\rceil \times efSearch_{L0} \times C \times T_{DC_{slow\_mem}}$$

$$\leq search\_time\_constraint$$

(1)

**Satisfy both response time and accuracy constraint.** Beyond response time constraint, high accuracy is clearly also important for high-quality ANNS, because otherwise users will not be able to find what they are looking for. In practice, the accuracy of search must be higher than an accuracy target $\theta$. Therefore, for a given HM-ANN graph, HM-ANN first applies Equation 1 to analytically get a set of candidate $(efSearch_{L0}, efSearch_{L1})$ pairs that satisfy the response time constraint. This step often significantly reduces the search space to only a small set of configurations.

Among those candidate pairs, HM-ANN uses a learning query set randomly sampled to measure the expected accuracy $\mathbb{E}(\theta)$, with $efSearch_{L1} \geq 1$, and $efSearch_{L0} \geq 0$ as constraints. HM-ANN then chooses those configurations that satisfy $\mathbb{E}[\theta] \geq \theta$. Finally, HM-ANN uses grid search to choose the configuration that leads to the shortest query time.

### 3.4 Complexity Analysis

**Search complexity.** HM-ANN constructs each layer as a navigable small world graph, which enables the number of hops scales logarithmically on the greedy search path. Similar to HNSW, HM-ANN constructs the graph with a fixed maximum number of links for each element, which guarantees that the average degree of each element in one layer is constant. The overall number of distance computation is proportional to a product of the number of hops and the average degree of the elements on the greedy path. Therefore, the search complexity in each layer of HM-ANN is logarithmic. Given a layer $i$ with $N_i$ elements, the search complexity of the layer $i$ is $O(log(N_i))$. Even with the bottom-up promotion, the maximum number of elements in each layer of HM-ANN remains $N$. Therefore, the overall search complexity of HM-ANN stays at $O(log(N))$.

**Index construction complexity.** The construction of HM-ANN contains two passes over the dataset, due to the top-down insertions and the bottom-up promotions. The insertion of an element involves a graph traversal followed by a constant cost of inserting short-range and long-range links. Therefore, this phase has a cost of $O(Nlog(N))$. The second pass of HM-ANN involves degree calculation and ranking and then extracts elements with high-degree in L0 into upper layers. Calculating the degree of all elements and sorting them in terms of the degree at L0 is bounded by $O(N \times M + Nlog(N))$. Therefore, in total the construction complexity of HM-ANN is $O(N \times M + Nlog(N))$.

**Memory usage complexity.** HM-ANN stores connection and elements separately in slow and fast memories. In particular, HM-ANN stores the connections in L0 and the elements that only appear in L0 into slow memory, and stores connections and elements at upper layers into fast memory. The fast memory consumption of HM-ANN equals to the sum of memory consumption of each layer (except L0): $fast\_memory\_size = \sum_{i=1}^{l}(N_i \times M_i) \times byte\_per\_link + N_1 \times byte\_per\_element$, where $N_i$ is the number of elements in layer $i$ ($i > 0$), and $M_i$ is the number of maximum established connections for each element in the layer $i$. The slow memory stores most of L0, which equals to $slow\_memory\_size = (N_0 \times M_0) \times byte\_per\_link + (N_0 - N_1) \times byte\_per\_element$.

## 4 Evaluation

### 4.1 Methodology

**Testing bed.** All experiments are done on a machine with Intel Xeon Gold 6252 CPU@2.3GHz. It uses DDR4 (96GB) as fast memory and Optane DC PMM (1.5TB) as slow memory.

**Workloads.** We use five datasets, BIGANN [22], DEEP1B [6], SIFT1M [22], DEEP1M [6], and GIST1M [4]. BIGANN contains one billion of 128-dimensional SIFT descriptors as a base set and 10,000 query vectors. DEEP1B contains one billion of 96-dimensional feature vectors of natural images and 10,000 queries. SIFT1M and DEEP1M are one-million subset vectors in BIGANN and DEEP1B respectively. GIST1M contains one-million 960-dimensional image descriptors.

**Table 1:** Indexing time and memory consumption for graph-based methods on billion-scale datasets

| | BigANN | | | | | DEEP1B | | | | |
|---|---|---|---|---|---|---|---|---|---|---|
| | Indexing | | | Search | | Indexing | | | Search | |
| | Graph size | Indexing time | Promo. rate | Fast-mem usage | Slow-mem usage | Graph size | Indexing time | Promo. rate | Fast-mem usage | Slow-mem usage |
| **HNSW** | 475GB | 90h | 0.02 | 96GB (hw caching) | 490GB | 723GB | 108h | 0.02 | 96GB (hw caching) | 748GB |
| **NSG** | 285GB | 115h | - | 96GB (hw caching) | 303GB | 580GB | 134h | - | 96GB (hw caching) | 599GB |
| **HM-ANN** | 536GB | 96h | 0.16 | 96GB | 462GB | 756GB | 117h | 0.11 | 96GB | 681GB |

**Evaluation metrics.** We measure the query response time as the average time of per-query execution time. We measure the accuracy with top-K recall (e.g., K=1, or 100), which measures the fraction of the top-K retrieved by the ANNS that are exact nearest neighbors.

**Comparison configurations.** For billion-scale tests, we include the following schemes: two state-of-the-art billion-scale quantization-based methods (IMI+OPQ [12] and L&C [13]); and the state-of-the-art non-compression-based methods (HNSW [29] and NSG [16]). To the best our knowledge, directly running HNSW and NSG at billion-scale points would trigger the out-of-memory error, and no prior work has been able to run HNSW and NSG with the two billion-scale datasets on a single machine, without compression. We therefore create two baseline configurations for both HNSW and NSG, using existing system-level data placement solutions: a *first-touch NUMA* configuration that places data in fast memory first until it is full and then in slow memory, and a *Memory Mode* configuration that treats fast memory as a hardware-managed fully-associative cache of slow memory. We include comparisons of HM-ANN at million-scale datasets with with HNSW [29] and NSG [16], which are known to be the best-in-class solution on the three million-scale datasets.

### 4.2 Experiment Results

**Billion-scale algorithm comparison.** We compare HM-ANN with the graph- (HNSW and NSG) and quantization-based algorithms (IMI+OPQ and L&C). For HNSW, we build graphs with $efConstruction$ and $M$ set to 200 and 48 respectively; For NSG we first build a 100-NN graph using Faiss [1] and then build NSG graphs with R = 128, L = 70 and C = 500. We collect results on NSG and HNSW using Memory Mode, since it leads to overall better performance than using first-touch NUMA (see Section 4.3 for the comparison of the two). For IMI+OPQ, we build indexes with 64- and 80-byte code-books on BIGANN and DEEP1B respectively. We present the best search result with search parameters nprobe=128 and ht=30 for BIGANN and with autotuning parameter sweep on DEEP1B. For L&C, we use 6 as the number of links on the base level, and use 36- and 144-byte OPQ code-books. We use the same parameters ($efConstruction$=200 and $M$=48) as HNSW to construct HM-ANN. We set $efSearch_{L0}$=2 and vary $efSearch_{L1}$ to show the latency-vs-recall trade-offs.

Figures 1 (a)-(d) visualize the results. Overall, HM-ANN provides the best latency-vs-recall performance. Figure 1 (a) and (b) show that HM-ANN achieves the top-1 recall of $> 95\%$ within 1ms, which is 2x and 5.8x faster than HNSW and NSG to achieve the same recall target respectively. IMI+OPQ and L&C cannot reach the similar recall target, because of precision loss from quantization. As another point of reference, the SSD-based solution, DiskANN [40] (not open-sourced), provides 95% top-1 recall in 3.5ms. In contrast, HM-ANN provides the same recall in less than 1ms, which is at least $3.5\times$ faster. We compare top-100 recall shown in Figures 1 (c) and (d). HM-ANN provides higher performance than all other approaches. For example, it obtains top-100 recall of $> 90\%$ within 4 ms, while performs 2.8x and 5x faster than HNSW and NSG with the same recall target respectively. Quantization-based algorithms perform poorly and have difficulties to reach a top-100 recall of 30%.

Table 1 shows the index construction time and index size of HNSW, NSG, and HM-ANN. Among the three, HNSW takes the shortest time to build the graph. HM-ANN takes 8% longer time than HNSW, because it takes an additional pass for the bottom-up promotion. However, HM-ANN is still faster to construct than NSG. In terms of memory usage, HM-ANN indexes are 5–13% larger than HSNW, because it promotes more nodes from L0 to L1. In terms of memory usage, HM-ANN consumes less fast memory than HNSW and NSG, which is valuable to reduce production cost [30, 38]. HNSW and NSG use all fast memory because they do not explicitly manage HM and by default using Memory Mode consumes all fast memory. The sum of slow and fast memory consumption can be larger than the index size, because there are metadata needed for search that are not counted into the index size.

**Million-scale algorithm comparison.** Besides the billion-scale tests, we evaluate HNSW, NSG and HM-ANN with the three million-scale datasets, which can fit in DRAM. For HNSW and HM-ANN,

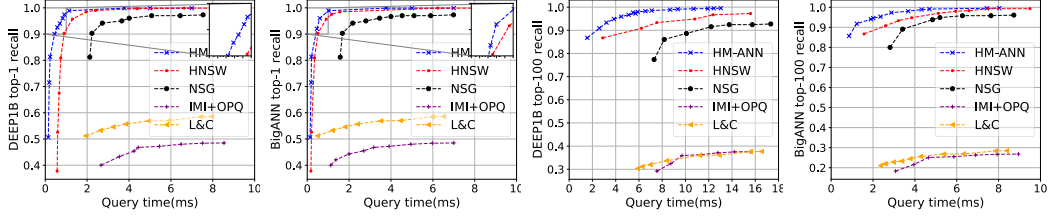

**Figure 1:** Query time vs. recall curve in (a) DEEP1B top-1, (b) BigANN top-1, (c) DEEP1B top-100, (b) BigANN top-100, respectively.

we set $efConstruction$ and $M$ to 100 and 16 for SIFT1M and DEEP1M; We set $efConstruction$ and $M$ to 100 and 32 for GIST1M. For NSG we use parameters in [2] suggested by the authors to build the graph. Figure 2 shows the result. Overall, HM-ANN achieves competitive and sometimes even better performance as HNSW and outperforms NSG on all three million-scale datasets. We further verify that the total number of distance computation from HM-ANN is lower (on average 850/query) than that of HNSW (on average 900/query) to achieve 99% recall target. This indicates that HM-ANN provides better accuracy-vs-latency results even when the datasets can fit in DRAM.

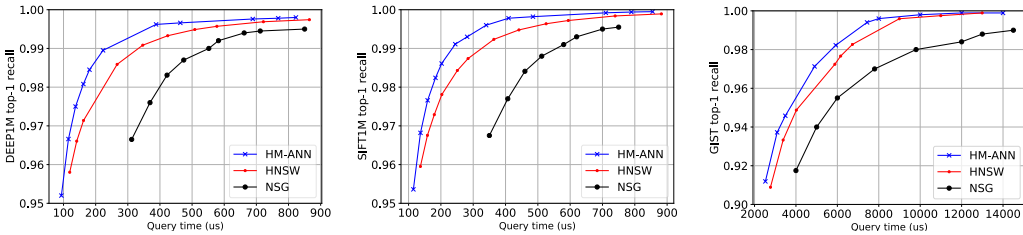

**Figure 2:** Query time vs. recall curve with (a)DEEP1M, (b)SIFT1M, and (c)GIST respectively.

## 4.3 Ablation Studies

**Effectiveness of high-degree promotion.** We compare the random promotion and high-degree promotion strategies. In this study, both strategies use the same number of promoted nodes for indexing and the same configurations for search. Figure 3 shows the results and indicates that high-degree promotion outperforms the baseline HNSW largely. The high-degree promotion performs 1.8x, 4.3x and 3.9x faster than the random promotion to reach 95%, 99%, and 99.5% recall targets, respectively, indicating that promoting high-degree nodes is effective for improving search efficiency.

$T_{fast\_mem}$ **and** $T_{slow\_mem}$ are measured by performing 10k distance computation in fast and slow memories and then report the average. $T_{slow\_mem}$ and $T_{fast\_mem}$ are 421ns and 183ns respectively.

**HNSW with Parallel L0 search.** We investigate whether it is sufficient to just modify the search procedure without modifying the hierarchical NN graph of HNSW to achieve similar performance gains as HN-ANN. Figure 4 shows the latency-vs-recall performance of default HNSW using parallel L0 search. We use $T$ nearest neighbours found during HNSW L1 search as entry points for the parallel search in L0, where $T$ is the number of parallel threads. We set $T = 4$, same as HM-ANN. HNSW with parallel L0 search only slightly outperforms HNSW. This suggests that parallel L0 search alone is not sufficient for performance improvement. Without it, the elements of L1 in HNSW are selected randomly and sparse, and the entry nodes found through L1 search are sub-optimal. As a result, even though the parallel search in L0 searches more nodes under the same time, the accuracy only slightly improves.

**Performance benefit of memory management techniques in HM-ANN.** Figure 5 contains a series of "stepping stones" between HNSW and HM-ANN to show how each optimization of HM-ANN contributes to its improvements. "HNSW + Bottom-up promotion (BP)" modifies the HNSW algorithm, mapping the bottom-most layer (i.e., L0) to the slow memory while building a high-quality projection of L0 in fast memory without significantly impacting search efficiency. It provides the benefit of improved search quality in fast memory while providing better entry points to L0 search in slow memory. Together with the parallel L0 search (i.e., "HNSW + Bottom-up promotion (BP) + Parallel L0 search (PL0)") it significantly improves the search efficiency versus running HNSW on HM without explicit data management. For example, to reach a 99% recall target, HM-ANN reduces

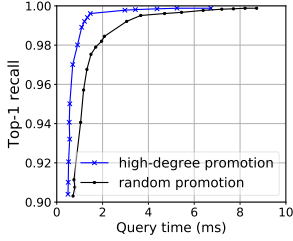

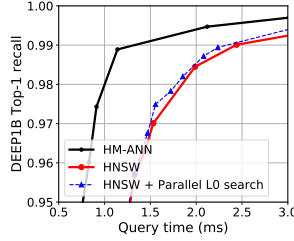

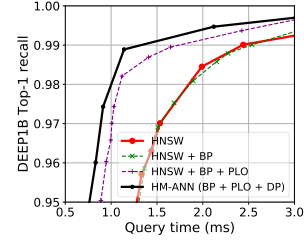

**Figure 3:** Comparison of two promotion strategies.

**Figure 4:** HNSW with parallel L0 search.

**Figure 5:** Comparison of techniques in HM-ANN.

the query time by 1.75x compared with HNSW. Finally, by prefetching data from slow memory to fast memory, HM-ANN further pushes the search efficiency frontier.

**System level data management solutions.** We compare HM-ANN with HNSW in Memory Mode and first-touch NUMA (as a software-based solution to manage data placement in HM, HM-ANN does not work with Memory Mode and first-touch NUMA). We also evaluate HNSW on slow memory without using any DRAM. Figure 6 shows the result. The figure shows that HM-ANN outperforms HNSW with Memory Mode and first-touch NUMA by 2x and 3.7x while achieving top-1 recall above 95%. The results suggest that although HM enables large memory capacity, simply using a system-level solution without algorithm change cannot make the best use of HM. Explicitly managing data for HM as HM-ANN does is the key to achieve superior latency and recall results.

**Effectiveness of performance model-guided search.** Figure 7 shows the distribution of $(efSearch_{l0}, efSearch_{l1})$ pairs that meet time constraint of <1ms and recall constraint of ≥90%. The bottom-left and top-right regions include those pairs violating either recall or time constraint; The colored regions are those meeting the constraints; The darker color has shorter query time. Figure 7 shows the performance model removes most of configurations violating the constraints.

To show effectiveness of performance modeling, we evaluate HM-ANN with BIGANN and 5 latency constraints from 1ms to 5ms (vertical red lines) in Figure 8. Red triangles represent $(efSearch_{l0}, efSearch_{l1})$ that meet the latency constraints set by Eqn. 1. Among those, we list 9 recall constraints marked with horizontal blue lines. For those recall constraints, 9 five-stars are those selected by HM-ANN, which meet the corresponding recall constraints while also having the shortest query time.

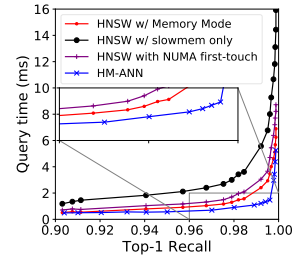

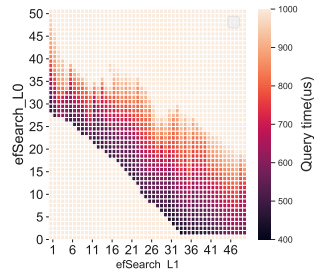

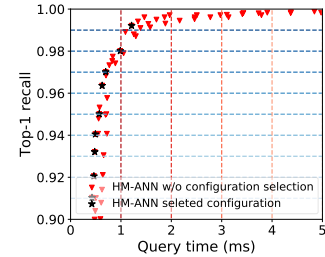

**Figure 6:** Comparison of data management methods.

**Figure 7:** Distribution of $efSearch$.

**Figure 8:** Performance with various $efSearch$.

## 5   Conclusions

HM can store billions of point database in a single machine. However, indexing and search algorithms on HM must be re-designed to release large performance potential of HM. We present a new graph-based indexing and search algorithm called HM-ANN, which maps the hierarchical design of the graph-based ANNs with memory heterogeneity in HM. Furthermore, HM-ANN adjusts the amount of distance computations at different layers to allow most accesses happen in upper layers stored in fast memory. Combined with a set of system-level techniques, HM-ANN is able to avoid expensive accesses in slow memory without sacrificing accuracy. Evaluation on billion-scale datasets show that HM-ANN establishes the new state-of-the-art for indexing and searching billion point datasets.

## Broader Impact

In this paper, we introduce HM-ANN, a hierarchical graph-based similarity search algorithm to leverage emerging heterogeneous memory, aiming to serve extremely large-scale data points on a single node with high accuracy and ultra fast response time.

The similarity search algorithm has been applied to a wide range of applications, including large-scale image/text search, web question and answer, and recommendation systems. Our research could be used to improve quality of services for those applications, increasing system scales without adding too much production cost, and efficiently handling large data sets with increasing volumes in data centers.

Furthermore, HM is an emerging architecture providing extremely large memory capacity for data intensive applications. Our research reveals a new field that could benefit from this architecture and shows great potential of using HM to establish the new state-of-the-art for indexing and searching large-scale datasets. Other algorithms that have the similar workload characteristics as ANN (such as the hierarchical design in ANN) can benefit from our research.

Although there are many benefits of using HM-ANN, we must pay attention to the potential risks of HM-ANN. HM-ANN uses a highly-structured approach to build the graph and removes randomness during node promotion. Although this approach is necessary to improve search quality and manage memory accesses in slow memory, it raises a risk of explicitly exposing critical information (such as nodes with high degrees) into specific memory regions, allowing the hacker to steal the information from the structured graph. Furthermore, Optane-based HM, which is one of the most common HM, raises security issues because of using non-volatile memory (i.e., Optane) [48, 5, 39]. Those security issues could happen in ANN search based on the Optane-based HM.

To mitigate the risks associated with using HM for ANN, we encourage research to understand the impacts of using HM-ANN in real-world scenarios and consider how the system (especially the address mapping scheme and memory organization) should be evolved to introduce randomness to mitigate security risk. More fundamentally, how to strike a good balance between high performance using a structured approach and potential security issues should be considered more broadly.

## Acknowledgments and Disclosure of Funding

This work was partially supported by U.S. National Science Foundation (CNS-1617967, CCF-1553645 and CCF-1718194).

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
