[Supplementary Material · NeurIPS_20__HMANN_Appendix.pdf]

# Appendix

**HM-ANN Graph Search Algorithm**    Algorithm 3 depicts the search algorithm in HM-ANN. The search begins at the entry point in the top layer (Line 1 in Algorithm 3) and then performs a one-greedy search from the layer $i$ ($i > 2$) to the layer 2 (Lines 4 to 6 in Algorithm 3). To obtain high quality search, HM-ANN performs beam search in L1 and L0 with search budget length $efSearch_{L1}$ (Line 7 in Algorithm 3) and $efSearch_{L0}$ (Line 8 in Algorithm 3) respectively. Different from searching in the upper layers, HM-ANN uses multiple entry points in L0. The number of entry points in L0 equals to $efSearch_{L1}$.

---

**Algorithm 3: HM-ANN K-NN-Search**

---

**Input:** multi-layer graph HM-ANN, query element $q$, number of nearest neighbour to return $K$
**Output:** K nearest elements to q
**Parameters:** size of the dynamic candidate list in layer 1 and 0 as $efSearch_{l1}$ and $efSearch_{l0}$ respectively

1   $ep \leftarrow$ entry point of HM-ANN;
2   $L \leftarrow$ level of $ep$ ;
3   $W \leftarrow \varnothing$;
4   **for** $i \leftarrow L...2$ **do**
5   $\quad$ $W \leftarrow$ search_layer($q, \{ep\}, ef = 1, i$);
6   $\quad$ $ep \leftarrow$ get nearest element from $W$ to $q$;
7   $W \leftarrow$ search_layer($q, \{ep\}, efSearch_{l1}, 1$);
8   $W \leftarrow$ search_layer($q, W, efSearch_{l0}, 0$);
9   **return** K nearest elements from $W$ to $q$

---