[Reviews · NeurIPS 2020]

Review 1

Summary and Contributions: it presents a novel graph-based similarity search algorithm called HM-ANN, which takes both memory and data heterogeneity into consideration and enables billion-scale similarity search on a single node without using compression.

Strengths: This paper is easy to follow and achieves the SOTA results. And the iead is to make sense.

Weaknesses: The top-1 recall in datasets are limited than other methods.

Correctness: yes

Clarity: yes

Relation to Prior Work: yes

Reproducibility: Yes

Additional Feedback:


Review 2

Summary and Contributions: This paper studies the problem of approximate nearest neighbor search (ANNS) and aims at redesigning the state-of-the-art solution towards the emergence of new hardware architecture, i.e. heterogeneous memory (HM). HM has two levels, with the first level is the traditional faster and smaller memory and the second level is a new slower and larger memory. Compared with traditional memory, HM offers more memory space with the new second level. As a tradeoff, HM introduces data fetching cost that frequently moves data from the second level to the first level. The paper claims four contributions. First, it presents the first billion-scale ANNS using HM. Second, it optimizes data fetching to improve ANNS query speed. Third, it proposes a performance model to automatically set hyperparameters for ANNS according to practical time and accuracy constraints. Fourth, it demonstrates that the proposed algorithm outperforms the state-of-the-art HNSW and NSG algorithms.

Strengths: S1. The paper introduces a new algorithm HM-ANN that redesigns the state-of-the-art ANNS algorithm towards heterogeneous memory. S2. The paper presents the performance of billion-scale ANNS that does not compress data points. S3. The proposed algorithm HM-ANN achieves better query efficiency than the state-of-the-art algorithms, including HNSW and NSG. Previous billion-scale ANNS algorithms mostly compress data points due to memory insufficiency. This work perceives the emergence of hierarchical memory (HM) and explores ANNS on uncompressed data. The work further proposes a new algorithm HM-ANN that optimizes data fetching in HM to improve query efficiency. The authors conducted extensive experiments to report the performance of billion-scale ANNS. The experiments demonstrate that HM-ANN outperforms the state-of-the-art HNSW and NSG algorithms and are 2X to 5.8X faster. [Author Response] The authors have addressed some of the comments in the response. I am happy to accept this paper.

Weaknesses: W1. The paper makes an inaccurate claim about the presence of billion-scale ANNS solutions. W2. The performance gain of the proposed HM-ANN algorithm seems marginal when considering its learning curve in practice. W3. The experiments do not evaluate the performance of data fetching. So it is hard to conclude that the proposed HM-ANN achieves better utilization of HM. The paper claims that the proposed HM-ANN is the first billion-scale ANNS solution on a single machine, without using compression (see the last paragraph of Introduction). The claim is questionable since the paper presents four existing solutions, i.e. HNSW, NSG, IMI+OPQ, and L&C. These existing solutions seem applicable to billion-scale ANNS, single machine, no compression, according to Figure 1(a) – (d). It seems each solution can process a billion uncompressed data when using Hierarchical Memory. The authors may consider to change the claim or make more explanations. Besides, the query efficiency improvements of HM-ANN seem marginal. In Figure 1(a), when achieving a recall of 0.9, HM-ANN is about 1 time faster than HNSW. In Figure 1(b), when achieving a recall of 0.9, HM-ANN is not clearly faster than HNSW. HM-ANN achieves the largest improvement in Figure 1(c). At recall 0.9, the runtime is about 2 times faster than HNSW. Overall, HM-ANN does not clearly outperform HNSW in many experiments. Given HM-ANN costs a lot of human efforts in learning curve and deployment, it is unclear how likely practitioners will use HM-ANN to replace HMSW. Although the paper claims HM-ANN is superior in data fetching optimizations, there are no experiments verifying the claim. The experiments report index construction time, index size, and query time. But none of the metrics directly related to data fetching efficiency. It is too early to conclude that the better efficiency of HM-ANN comes from data fetching optimizations. Although the paper presents many details on the optimizations, there are no theoretical justifications or experiments to evaluate the optimizations.

Correctness: Please refer to Weaknesses.

Clarity: The paper is well written except for potentially questionable claims. Please refer to Weaknesses.

Relation to Prior Work: I do not find discussions in this work that mention prior contributions.

Reproducibility: Yes

Additional Feedback:


Review 3

Summary and Contributions: This paper introduces an adapted version of the HNSW graph-based approximate nearest neighbor search algorithm. It is adapted to PMM, a storage medium that is between RAM and Solid State Disk (SSD) in terms of speed-capacity tradeoff. The changes consists in: - storing the lower level L0 of the HNSW on PMM, the rest remains in RAM, hence the heterogeneous memory (HM) name. - a better selection of the L0 points to put in L1 based on the node degrees - doing a beam search in L1 as opposed to a greedy search

Strengths: The results are truly impressive. It is impossible to obtain such high recalls with in-RAM compression. The comparisons are fair, especially because the authors did a good job to explore more straightforward ways to exploit HM (and show they are inferior).

Weaknesses: The hardware setup (Heterogeneous Memory) is uncommon, which makes this a niche application The algorithmic adaptations are somewhat straightforward

Correctness: The method is simple, so there is not much room for incorrectness. The discussion of the "performance model" in 3.3 is a bit disappointing because after the derivation it just boils down to a grid search on the few parameters of the method, which is what most other papers do as well.

Clarity: The paper is very well written overall. Two points: - in the beginning of sec 3 it is not clear to me what the difference is between the degree of a node and the nb of established connections in the HNSW data structure. My understanding is that the degree is the number of incoming edges in a K-NN graph. - please avoid long math identifiers like "efSearch_L0" : they look more like variable identifiers in computer code - it would be useful to recap what properties of the slow-memory make this possible: is it fast memory access? Throughput? threaded access? Could it be applied to SSD storage as well?

Relation to Prior Work: The prior work for billion-scale indexing is relevant. The baselines are probably slightly outdated, since inverted-index based methods now often use a HNSW structure as top-level quantizer (see eg. https://github.com/dbaranchuk/ivf-hnsw or the Faiss recommended indexes). This should not invalidate the main results of the paper though.

Reproducibility: Yes

Additional Feedback: *** after rebuttal: The authors have provided a serious rebuttal. I particularly appreciate the 2nd item that decomposes the contribution of each improvement. I would suggest the authors incorporate it in a final version. ***


Review 4

Summary and Contributions: This paper studies approximate nearest neighbour (NN) search and adapts the popular HNSW [28] algorithm with hierarchical heterogeneous memory support to query billion-scale datasets. The key adaptations are: (1) promoting elements with the highest degrees to upper layers in the hierarchical NN graph structure, (2) promoting more elements to the upper layers (to take advantage of the fast search efficiency of the hardware used to store such elements), and (3) parallel L0 search with multiple entry points in L0. A cost model is designed to help algorithm parameter selection. Experimental results on both billion-scale and million-scale datasets confirm the effectiveness and efficiency of the proposed algorithm. ***Update after author rebuttal: I appreciate the authors' efforts to address my comments. I am happy with the rebuttal and am updating my score from "Marginally below the acceptance threshold" to "Marginally above the acceptance threshold".

Strengths: 1. An important problem is studied -- approximate nearest neighbour search has wide applications in data mining and database problems. 2. A cost model is designed to help algorithm parameter selection. 3. Experimental results on large-scale real data confirm the effectiveness and efficiency of the proposed algorithm.

Weaknesses: The proposal is a simple and effective adaptation of the HNSW algorithm that yields fast and accurate ANN search results on billion-scale data. The main concern is that the technical depth of the paper seems somewhat limited. There is no theoretical analysis on the approximation ratio of the proposed algorithm, or the approximation ratio in comparison with HNSW due to the proposed adaptations. The results may suit a less theoretical venue. An additional baseline is needed: adapting HNSW by adding a parallel search to L0 with multiple entry points from L1 (similar to what Lines 8 and 9 in Algorithm 3 do). This will help demonstrate whether it is sufficient to just modify the search procedure (without modifying the hierarchical NN graph of HNSW) to achieve similar performance gains to what HM-ANN yields.

Correctness: Line 316: "Figure 2 shows the result. Overall, HM-ANN achieves competitive and sometimes even better performance as HNSW." This seems to contradict with the figure where HM-ANN outperforms HNSW in most cases. Line 297: "DiskANN [38] (not open-sourced), provides 298 95% top-1 recall in 3.5ms. In contrast, HM-ANN provides the same recall in less than 1ms". Are these numbers obtained from the same hardware and software settings?

Clarity: The paper is in general very well written and easy to follow. Adding a figure to illustrate the modified bottom-up promotion procedure may help further improve the readability of the paper.

Relation to Prior Work: The introduction and related work sections contain sufficient details on related studies. Section 3 contains sufficient details on how the proposed technique HM-ANN differs from HNSW.

Reproducibility: Yes

Additional Feedback: See above. Line 3 of Algorithm 1: v -> $v$ PS. This is an emergency review called upon two days ago and hence it was a bit late in the submission.

[Author Response · NeurIPS 2020]



Figure 1: Comparison of techniques  Figure 2: Search perf. on BigANN.  Figure 3: HNSW with parallel L0 search.

**The contribution of HM-ANN on billion-scale ANNS. (R2)**  The core of our work is to show that we can host billion-scale datasets on a single machine with *both* high accuracy and fast speed using HM, outperforming existing solutions. We will revise the claim in the paper to a more accurate one as "a fast and accurate billion-scale nearest neighbor search solution on a single node without compression".

**The improvement from data prefetching from slow memory to fast memory. (R2)** Figure 1 contains a series of "stepping stones" between HNSW and HM-ANN to show how each optimization of HM-ANN contributes to its improvements, including the data prefetching (DP) asked by the reviewer. "HNSW + Bottom-up promotion (BP)" modifies the HNSW algorithm, mapping the bottom-most layer (i.e., L0) to the slow memory while building a high-quality projection of L0 in fast memory without significantly impacting search efficiency. It provides the benefit of improved search quality in fast memory while providing better entry points to L0 search in slow memory. Together with the parallel L0 search (i.e., "HNSW + Bottom-up promotion (BP) + Parallel L0 search (PL0)") it significantly improves the search efficiency versus running HNSW on HM without explicit data management. For example, to reach a 99% recall target, HM-ANN reduces the query time by 1.75x compared with HNSW. Finally, by prefetching data from slow memory to fast memory, HM-ANN further pushes the search efficiency frontier.

**Deployment effort of HM-ANN. (R2)** Our performance model has the benefit of significantly pruning the parameter search space that cannot satisfy the response time and accuracy constraints, so it actually expedites the deployment.

**The performance gain of HM-ANN on billion-scale datasets. (R2)** If we look at the high accuracy range, HM-ANN obtains top-1 recall of >95% and >99% within 0.5 ms and 1.3 ms respectively, which is 2x and 3.3x faster than HNSW. This can be seen from the latency-vs-recall curve with recall larger than 90% for the BIGANN1B dataset in Figure 2.

**HM-ANN and SSD storage. (R3)** HM-ANN tries to access the data in slow memory, which means HM-ANN does not work with SSD-based storage directly. Combined with SSD, HM-ANN can be used as an in-memory index for hosting even larger datasets. This would make an interesting future study.

**Theoretical analysis on the approximation ratio. (R4)** We agree it is important to have a theoretical guarantee in terms of the approximation ratio. However, we expect it to be difficult, since all the existing state-of-the-art graph-based NNS algorithms (e.g., HNSW) lack such guarantees. There are multiple paths to advance the state-of-the-art ANN search, and this work focuses on one of them. We can still at least provide a more detailed search time complexity analysis for HM-ANN. HM-ANN constructs each layer as a navigable small world graph, which enables the number of hops scales logarithmically on the greedy search path. Similar to HNSW, HM-ANN constructs the graph with a fixed maximum number of links for each element, which guarantees that the average degree of each element in one layer is constant. The overall number of distance computation is therefore proportional to a product of the number of hops and the average degree of the elements on the greedy path. Therefore, the search complexity in each layer of HM-ANN is logarithmic. Given a layer $i$ with $N_i$ elements, the search complexity of the layer $i$ is $O(log(N_i))$. We then analyze the overall search complexity of HM-ANN. Even with the bottom-up promotion, the maximum number of elements in each layer of HM-ANN remains $N$. Therefore, the overall search complexity of HM-ANN stays at $O(log(N))$.

**HNSW with Parallel L0 search. (R4)** We added an experiment to investigate whether it is sufficient to just modify the search procedure without modifying the hierarchical NN graph of HNSW to achieve similar performance gains as HN-ANN. Figure 3 shows the latency-vs-recall performance of default HNSW using parallel L0 search. We use $T$ nearest neighbours found during HNSW L1 search as entry points for the parallel search in L0, where $T$ is the number of parallel threads. We set $T = 4$, same as HM-ANN. HNSW with parallel L0 search only slightly outperforms HNSW. This suggests that parallel L0 search alone is not sufficient for performance improvement. Without it, the elements of L1 in HNSW are selected randomly and sparse, and the entry nodes found through L1 search are sub-optimal. As a result, even though the parallel search in L0 searches more nodes under the same time, the accuracy only slightly improves.

**DiskANN evaluation hardware and software settings. (R4)** The evaluation platform of HM-ANN and that of DiskANN have the same CPU, and the memory bandwidth and latency of two evaluation platforms are comparable. The software settings are also similar.

[Meta-Review · NeurIPS 2020]

The paper attempts to scale nearest neighbor search using heterogenous memory hardware. In this regard, authors devised a practical trick on top of HNSW. It is a clean node promotion strategy along the memory hierarchy using the degree information. The method was evaluated on some common large datasets, but not necessarily difficult ones. Reviewers found the setup to leverage the memory hierarchy interesting and the benefits obtained from it appears promising. Although all reviewers agree that algorithmic contribution is straightforward and fairly limited, but simplicity is not always bad. The other concern was whether the empirical evaluation is sufficient and as a result an extra expert review was sought after the feedback period (which is provided below in detail). In particular, there are still a few missing comparison to baselines and tuning system based caching properly. Authors are strongly encouraged to add these further experiments to the final version of the paper. Overall owing to simplicity and niche application, I am recommending acceptance to NeurIPS. Another expert review === 4:reject === Overview: The central contribution of the algorithm is a graph node promotion strategy, and this is applied to HNSW algorithm to perform fast nearest neighbor search. The node promotion strategy basically moves large-degree graph nodes higher in the memory hierarchy. In the ablation test, this simple strategy is compared against random node promotion, or using a system-level data management solution, and the paper showed the proposed strategy works better in terms of latency at higher recall region. === Pro: 1. Working on a practically useful direction leveraging cutting-edge hardware for fast nearest neighbor search. 2. Provided source code. Cons: 1. The amount of algorithimic innovation is pretty small. While node promotion based on degree is a simple and nice strategy, this is effecitvely a heuristic to cache more-frequently accessed items. As such, I am not very convinced that this is significant enough for a NeurIPS conference paper. In my opinion, there are other more important, unsolved problem in this architecture. For example, the indexing time of 1B nodes requires 96/117hours, and this practically limits the ability of the algorithm to further scale up to larger datasets. 2. There is insufficient amount of comparison to DiskANN. There is no big difference between using SSD or Optane for NNS (except the underlying hardware is different). Implementation is can be very similar for both hardware and can be done through file IO or mmap-ed memory. While I understand DiskANN is not open source, it will be at least nice to test out the strategy in this paper on SSD and compare it against DiskANN's numbers. === Question: In Figure4, system based caching strategy "memory mode" is surprisingly effective, almost on-a-par with the proposed method at 90% recall. The proposed caching strategy becomes visibily more effective only after 98%. This is a little alarming to me and makes me feel it might be possible to "tune" the system strategy to achieve the more or less same effect of the proposed strategy. === Conclusion: While this paper takes a practically interesting direction, the amount of research contribution seems incremental and some important experiments are missing (comparison to DiskANN).